



**A refinement of coccolith separation methods: Measuring the sinking**
**characters of coccoliths**
Hongrui Zhang[1, 2], Heather Stoll[2], Clara Bolton[3], Xiaobo Jin[1], Chuanlian Liu[1]
[1] State Key Laboratory of Marine Geology, Tongji University, Shanghai, 200092, China
[2] Geological Institute, Department of Earth Science, Sonneggstrasse 5, ETH, 8092, Zürich, Switzerland
[3] Aix-Marseille Univ, CNRS, IRD, Coll de France, CEREGE, Aix en Provence, France.
*Correspondence to:* Chuanlian Liu (liucl@tongji.edu.cn)
**Abstract.** The sinking velocities of individual coccoliths are relevant for export of their CaCO3
from the surface ocean, and for laboratory methods to separate coccoliths of different sizes and
species for geochemical analysis. In the laboratory, the repeat settling/decanting method was the
earliest method to separate coccolith from sediments for geochemical analyses, and is still widely
used. However, in the absence of estimates of settling velocity for non-spherical coccoliths, previous
implementations have depended mainly on time consuming empirical method development by trial
and error. In this study, the sinking velocities of coccoliths belonging to different species were
carefully measured in a series of settling experiments for the first time. Settling velocities of modern
coccoliths range from 0.154 to 10.67 cm h$^{-1}$. We found that a quadratic relationship between
coccolith length and sinking velocity fits well and coccolith sinking velocity can be estimated by
measuring the coccolith length and using the length-velocity factor, $k_{sv}$. We found a negligible
difference in sinking velocities measured in different vessels. However, an appropriate choice of
vessel must be made to avoid 'hindered settling' in coccolith separations. The experimental data and
theoretical calculations presented here will support and improve the repeat settling/decanting
method.



## 1. Introduction

Coccolithophores are some of the most important phytoplankton in the ocean. They can secrete calcareous plates called coccoliths, which contribute significantly to discrete particulate inorganic carbon in the euphotic zone and to $CaCO_3$ fluxes to the deep ocean (e.g., Young and Ziveri, 2000; Sprengel et al., 2002), and record paleoenvironmental changes (e.g., Beaufort et al., 1997; Stoll et al., 2002; Zhang et al., 2016). However, the use of coccolith geochemical analyses in paleoenvironmental reconstructions is hindered by the difficulty of isolating coccolith compared with foraminifera. Two main methods have been developed to concentrate near-monospecific assemblages of coccoliths from bulk sediments: one is the method based on a decanting technique (Paull and Thierstein, 1987; Stoll and Ziveri, 2002) and the other is that based on microfiltration (Minoletti et al., 2008). The improvement of separation techniques offered a new perspective to study the Earth's history (e.g. Stoll, 2005; Beltran et al., 2007; Bolton and Stoll, 2013; Rousselle et al., 2013; Tremblin et al, 2016). Moreover, the development of coccolith oxygen and carbon isotope studies in culture in recent years (e.g. Ziveri et al., 2003; Rickaby et al., 2010; Hermoso et al., 2016; McClelland et al., 2017) has provided an improved mechanistic understanding of coccolith isotope data and therefore stimulated the need for more purified coccolith fraction samples from the fossil record.

Both decanting and microfiltering are widely used methods for coccolith separation. Microfiltering relies heavily on the specifications of micro filter membrane (such as 2μm, 3μm, 5μm and 8μm pore size) and is highly effective in the large size range, but is very time consuming in sediments with a high proportion of very small coccoliths. It is also impossible to separate coccoliths with similar lengths by microfiltration, such as *Florisphaera profunda* and *Emiliania huxleyi* (Hermoso et al., 2015) Decanting, on the other hand, is highly effective for the small-sized coccoliths, because their slow settling times permit greater ability to separate different sizes. Consequently, in some studies, a combination of the micro filtering and sinking or centrifugation method were applied for coccolith separation (Stoll, 2005; Hermoso et al., 2015). The repeated sinking/decanting method, first employed by (Edwards, 1963; Paull and Thierstein, 1987) follows the simple principle formalized by Stokes' Law for spherical particles: particles of larger size settle more quickly because they have a higher ratio of volume and mass (accelerating sinking) to sectional





area (resistance retarding sinking).    However, the sinking velocities of coccoliths with complex
shape are difficult to calculate and have not been quantified in previous studies. Consequently, the
repeated decanting method has generally used settling times based on empirical trial and error.
In this current study, we present a novel and rigorous estimation of the sinking velocity for 16
species of modern and Cenozoic coccoliths, carefully measured in 0.2% ammonia at 20℃. With this
new dataset, we explore how to estimate the sinking velocity of coccoliths by shape and length,
which allows our estimations to be generalized for other species, and for situations where the mean
thickness of coccoliths of a given species was different from that of our study.    These
generalizations, together with our results on sinking velocities of one coccolith species
(*Gephyrocapsa oceanica*) in different vessels, should allow a significant improvement in efficiency
of future protocols for separation of coccoliths by repeated decanting.
**2. Materials and methods**
**2.1 Sample selections**
We measured the sinking velocity of 16 different species of coccoliths, principally of Quaternary
age but including two Neogene samples (Figure 1). Numbers of small coccoliths, including *E.*
*huxleyi*, *Gephyrocapsa* spp and *Reticulofenestra* spp. are about a magnitude greater than that of
larger coccoliths. However, the larger coccoliths' contributions to carbonate can be as high as 50%
(Baumann, 2004; Jin et al., 2016). Moreover, both small coccoliths and large coccoliths are useful
in geochemical analyses (Ziveri et al., 2003; Rickaby et al., 2010; Candelier et al., 2013; Bolton et
al., 2012, 2016; Bolton and Stoll, 2013). Therefore, both small and large coccoliths were studied in
this research. The coccoliths were isolated from eight samples from the Pacific and Atlantic Oceans
(more location information are in Figure 1 and Table A1; the pictures of studied coccolith can be
found in Appendix B). All classifications of coccolith follow Nannotax3 except *Reticulofenestra*
spp. (Figure C2 in Appendix C).
**2.2 Experiment designs**
**2.2.1 Sample pretreatments**
The sinking velocity measurement depends on absolute abundance estimation (more details in 2.2.2).
However, on microscope slides, larger coccoliths and foraminifer fragments may cover smaller





coccoliths, reducing the accuracy of coccolith absolute number. Thus, before sinking experiments
were carried out, raw sediments were pretreated to purify the target coccoliths to reduce errors in
coccolith counting. The raw sediments were disaggregated in 0.2% ammonia and sieved through a
63 μm sieve and then treated by sinking method or filtering method (Bolton et al., 2012; Minoletti
et al., 2008) to concentrate the target species up to at least more than 50% of total assemblages (for
Noëlaerhabdaceae coccoliths, a percentage more than 90% can be easily achieved). Most of species
were measured individually in settling experiments, except the *Pseudoemiliania lacunosa* and
*Umbilicosphaera sibogae*, which cannot be separated from each other.
**2.2.2 Measuring the sinking speeds of coccoliths**
We are not aware of any prior direct determination of the sinking velocity of individual coccoliths,
although the sinking velocities of live coccolithophores and other marine algae cells have been
successfully measured by the 'FlowCAM' method (Bach et al., 2012) or similar photography
technique (e.g. Miklasz and Denny, 2010). Here we introduce a simple method to measure the
particle sinking speeds without special equipment. After pretreatment, the coccolith suspensions
were gently shaken and then moved into comparison tubes which were vertically mounted on tube
shelves. We set the timer going and let the suspension settle for a specified period of time, marked
as sinking time or settling duration (T). Thereafter, we removed the upper 15 ml supernatant in a 50
ml centrifuge tube with a 10 ml pipette. This operation should be performed slowly and gently to
avoid drawing lower suspensions upward. The number of coccoliths in the upper and lower
suspensions were carefully counted by the 'drop technique', which is a quick method to determine
absolute abundance of coccoliths (Koch and Young, 2007; Bordiga et al., 2015).
To calculate the sinking velocities of coccoliths, we define a parameter named the separation ratio
(R), which represents the percentage of removed coccoliths in one separation. This parameter is
important and will be repeatedly mentioned in the following part. R was measured using the
following equation (more details about derivation can be found in Appendix D):
$$R = \frac{\frac{N_1}{n_1} \times V1}{\frac{N_1}{n_1} \times V1 + \frac{N_2}{n_2} \times V2} \qquad (2\text{-}1)$$
where N1 and N2 are numbers of coccoliths counted in upper and lower suspension slides,
respectively; n1 and n2 are the number of fields of view (FOV) counted. V1 and V2 are the volume
of the settling vessel defined by the settling distance, as shown in Figure 2.





The separation ratio, R, also has a relationship with sinking time, T:
$$R = \frac{V_1 \ - \frac{V_1}{D} \times sv \times T}{V_1 + V_2}$$
(2-2)

where V1, V2 and D are shape parameters shown in Figure 2; and sv is the average sinking velocity
of measured coccoliths. If we plot R against T, the slope of line has a relationship with sv. Hence
liner regressions between R and T were processed with MATLAB to calculate the sv (details about
error analyses can be found in Appendix E).
There are still two issues to be explained. The first one is to eliminate the shape differences among
vessels, all separation ratios have been transferred to calibrated separation ratios (Rcal), which
means the separation ratio measured in a standard vessel (more details in Appendix D). The other
one is that we treated the average sinking velocities as the sinking velocities of the coccoliths with
the average length. This approximation has been proved reasonable in Appendix D.
**2.2.3 Detecting the potential influence of vessels**
Seven commonly used vessels were selected to detect the potential influence of vessels (Figure 3).
Two of them are made of plastics (No.2 and No.3 in Figure 3) and all others are pyrex glass vessels.
About 500 mg of sediment from the core KX21-2 were pretreated as described in 2.2.1 and
suspended in about 500 ml ammonia. After that, settling experiments were performed as described
in 2.2.2 using different vessels. In these experiments, only the dominant species, *G. oceanica*, was
measured.
**2.2.4 Other factors influencing the sinking velocity**
Temperature can change the density and viscosity of liquid. Generally speaking, the higher the
temperature is, the lower the density and viscosity will become and the faster pellets will sink. Take
water for instance, if the temperature increases from 15 to 30℃, the particle sinking velocity will
increase by ~43% (Table 1). All sinking velocities measured or discussed in the following sections
were velocities at 20℃  to minimize the influence of temperature.
The calibration of sinking velocity in high concentration suspension has been calculated by
Richardson and Zaki (1954)
$$sv = sv_0 (1 - \alpha_s)^{2.7}$$
(2-3)

where the $\alpha_s$ is the solids volume fraction. Based on equation 2-3, the higher the suspension
concentration is, the slower the sinking velocity will be. That is so called 'hindered settling'. When





the $\alpha_s$=0.2%, the reduction of sinking velocity owing to hindred settling cannot be neglectable
(sv/sv$_0$ equals 99.46%). Hence, in this study all suspensions have solid volume fractions lower than
0.2% to avoid notable reduction of coccolith sinking velocities.

## 3. Results and Discussions

### 3.1 Influence of vessels

The sinking velocities of *G. oceanica* in the core KX21-2 in ammonia at 20℃  measured in different
vessels vary from 0.99 to 1.23 cm h$^{-1}$. The lowest value occurred in the 100 ml centrifuge tube and
the highest sinking velocity was measured in the 50 ml centrifuge tube experiments. The correlations
between sinking velocities and different vessel parameters are quite low: r=0.13 for the vessel inner
diameter, r=0.0005 for the sinking distance and r=0.051 for the upper volume and total volume ratio
(V1/(V1+V2)). The dissipation of energy by friction between the moving fluid and the walls can
cause a reduction of sinking speed (wall effect). A significant wall effect will be detected when a
particle is settling in a vessel which diameter is smaller than the particle size by two orders of
magnitude (Barnea and Mizarchi, 1973). The length of coccolith is on micron scales, so the
diameters of vessel used in laboratory are about more than three order of magnitude larger than
coccoliths. Moreover, our results show that the difference between vessel materials, glass and
plastics, can also be ignored (Figure 4). Hence, we suggest that vessel type almost has no significant
influence on sinking velocity of coccoliths.
However, our experiments were premised on the basis that the concentration of suspension was
equal among different vessels. This means that large vessels can treat more sediment at one time but
if we choose a larger vessel, more suspensions should be pumped and it often costs more time in
sinking (often due to longer sinking distance). Assuming that the sediment is composed of 50%
calcite (with density of 2.7 g cm$^{-3}$) and 50% clay (about 1.7 g cm$^{-3}$), the largest amount of sediment
that can be used without significant reduction of the sinking velocity (5%) is about 400 mg in 100
ml suspension (this calculation is based on equation 2-3). However, the sediments accumulating in
the lower suspension, the particle concentration can be more than 4 times higher than the initial
homogenous concentration. To avoid this, we recommend about 100 mg dry sediment should be
suspended in at least 100 ml suspension to avoid 'hindered settling'. If more sediment is necessary



for geochemistry analyses, then a larger vessel should be selected to separate enough sample in one
time.

**3.2 Sinking velocities at 20℃ in 0.2% ammonia**

We measured the separation ratios of different coccoliths in comparison tubes at 20℃ in 0.2%
ammonia (Figure 5). The sinking velocities of coccoliths were then calculated by linear fitting of
separation ratios and settling durations. The sinking velocities of studied coccoliths vary by one
order of magnitude from 0.154 cm h$^{-1}$ to 10.67 cm h$^{-1}$ (Table 2). The highest sinking velocity was
found in the measurement of *Coccolithus pelagicus* and the lowest velocity was found for *F.*
*profunda*. The average sinking speeds of coccolith is about 10-50% of the terminal sinking velocities
of calcite spheres calculated by Stokes' Law (Figure 6). These ratios are comparable with the oval
objects (e.g. seeds) data from Xie and Zhang (2001) and smaller than those from McNown and
Malaika (1950). The sinking velocities of coccoliths measured in our experiment are about 2-3
orders of magnitude smaller than values from sediment traps of 143-243 m d$^{-1}$ (595~1012 cm h$^{-1}$)
in the North Atlantic (Ziveri et al., 2000 and Stoll et al., 2007), confirming the fact that the coccoliths
sinking out of the euphotic layer are mainly in the form of sinking aggregates rather than individual
coccoliths.

**3.3 Estimating the sinking velocities**

Generally speaking, the sinking velocities of coccoliths increase with the distal shield length (Figure
5a), as expected from the increase in volume to sectional area for a given geometry as length
increases. Our data implies that the sinking velocity has a power function relationship with distal
shield length.
We propose that the sinking velocity of coccoliths might have a quadratic relationship with distal
shield length as described by Stokes' Law (Figure 6a). If we use data for all species except
*Helicosphaera carteri*, the sinking velocities can be described by the following equation:
$$sv = 0.0982\,(\pm 0.001)* \phi^2 \qquad (3\text{-}1)$$
Based on this quadratic regression, we derive a shape-velocity factor ($k_{sv}$) that relates settling
velocity to coccolith length.
$$sv = k_{sv}* \phi^2 \qquad (3\text{-}2)$$



Furthermore, this factor is analogous to the shape-mass factor, '$k_s$' used to relate coccolith mass to
coccolith length (Young and Ziveri, 2000). The length and shape-velocity factor of coccoliths can
be used to predict most of the sinking velocity variations, however, variations may also arise due to
changes in coccolith mass and thickness, for a given length, and due to the hydrodynamics of
particular shapes. We noticed that the smaller coccolith *G. caribbeanica* has a greater sinking
velocity than the larger coccolith, *G. oceanica*. We suggest that this was caused by greater mass per
length (or greater average thickness) in the case of *G. caribbeanica* and this may be due to the closed
central area while *G. oceanica* has an open central area. Another example is H. carteri, its smaller
sinking velocity can be explained by the unique structure: the broad edge of *H. carteri* can increase
the drag force significantly and *H. carteri* has the largest ellipticity (major axis length and minor
axis length ratio) among the measured coccoliths, which means the mass of *H. carteri* is smaller
than other species of coccoliths with similar lengths (Figure 6d and Figure C3). In the case of partial
dissolution, the well-preserved *Cyclicargolithu floridanus* may have higher mass than dissolved (or
disarticulated) *Cy. floridanus*, and therefore a slightly higher shape-velocity factor.
**4. Conclusions**
To improve coccolith separation by settling methods, we measured sinking velocities of different
coccoliths by gravity. Sinking velocities in this study varied from 0.154 to 10.61 cm h$^{-1}$, about 10%
to 50% of those of calcite spheres with same diameter. The shape of different vessels had little
impact on the sinking velocity. But we should consider the volume of vessels to avoid 'hindered
settling'. The sinking velocities are mainly controlled by the shape of coccolith, including the distal
shield length, the size of central area, and the ellipticity of coccoliths. Besides the shape of coccoliths,
temperature is also crucial to the coccolith separations because of the dependence of sinking
velocities on temperature.
Length-velocity factors were proposed to estimate coccoliths sinking velocities, so coccolith sinking
speeds in different samples can be easily estimated by following steps:
**1.** Measure the mean length of coccoliths under the microscope;
**2a.** For species which sinking speed has been directly measured, we can use the length-velocity
factor directly ($sv = k_{sv} * \phi^2$);
**2b.** For unmeasured species, we can choose the length-velocity factor of coccolith with similar



223 morphology in this study or use the general length-velocity formula (sv=0.098($\pm$0.001)* $\phi^2$)

224 If we use the general formula, it should be noted that a closed central area coccolith will sink faster

225 than prediction (for *G. caribbeanica* and small *C. leptoporus* will settle ~40% faster) and coccoliths

226 with greater ellipticity can settle much slower (for *H. carteri* will settle as 30% of the predicted

227 sinking velocity for coccolith with similar length).

228

229 *Acknowledgements.* This study was supported by grants from the Chinese National Science

230 Foundation (91428310, 91428309 and 41530964, to L.C.). We thank the Integrated Ocean Drilling

231 Program (IODP) for providing the samples. The IODP is sponsored by the U.S. National Science

232 Foundation and participating countries under management of the IODP Management International,

233 Inc (IODP-MI).



**Table 1.** The influence of temperature on sinking velocity. Density data is from Kell (1975) and

viscosity data is from Joseph et al. (1978).

| T (℃) | $\rho$ (g cm$^{-3}$) | $\eta$ (mPa s) | $SV_T : SV_{T=20}$ |
|---|---|---|---|
| 15 | 0.9991 | 1.1447 | 0.8804 |
| 20 | 0.9982 | 1.0087 | 1 |
| 25 | 0.9970 | 0.8949 | 1.1279 |
| 30 | 0.9956 | 0.8000 | 1.2627 |

**Table 2.** The sinking velocity and shape-velocity factor of different coccolith species: $\phi$ means the
distal shield length of coccolith and St $\phi$ is the standard deviation of distal shield length; sv represents
the sinking velocity; sv (95%-) and sv (95%+) represent the lower and higher limit of 95% confidence
level, respectively. '$k_{sv}$' represents the length-sinking velocity factor. The short name of coccolith can
be found in the caption of Figure 4. The details of coccoliths length distribution are in Appendix C.

| Species | abb. | $\phi$ (µm) | St $\phi$ (µm) | sinking velocity (cm h$^{-1}$) | Sv (95% -) | Sv (95% +) | $k_{sv}$ |
|---|---|---|---|---|---|---|---|
| *F. profunda* | Fp-WP | 1.508 | 0.557 | 0.158 | 0.010 | 0.011 | 0.070 |
| *F. profunda* | Fp-SCS | 1.786 | 0.641 | 0.154 | 0.051 | 0.052 | 0.048 |
| small *Reticulofenestra* | Ret (<4um) | 2.454 | 0.509 | 0.848 | 0.354 | 0.416 | 0.141 |
| *E. huxleyi* | Emi | 2.512 | 0.469 | 0.853 | 0.054 | 0.064 | 0.135 |
| *Gephyocapsa* spp. | G spp | 2.755 | 0.502 | 0.752 | 0.125 | 0.147 | 0.099 |
| *G. caribbeanica* | Gcar | 3.312 | 0.352 | 1.873 | 0.174 | 0.192 | 0.171 |
| *U. sibogae* | Umb | 4.060 | 0.500 | 1.268 | 0.416 | 0.441 | 0.077 |
| *G. oceanica* | Geo | 4.187 | 0.517 | 1.170 | 0.155 | 0.178 | 0.067 |
| *P. lacunosa* | Pla | 4.350 | 0.617 | 1.171 | 0.337 | 0.338 | 0.062 |
| Small *Ca. leptoporus* | Cal small | 4.605 | 0.629 | 3.351 | 0.172 | 0.199 | 0.158 |
| large *Reticulofenestra* | Ret(>4um) | 4.988 | 0.605 | 2.379 | 0.534 | 0.641 | 0.096 |
| *Cy. floridanus* | Cyf | 5.805 | 0.963 | 4.174 | 0.320 | 0.336 | 0.124 |
| (dissolved) *Cy. floridanus* | Cyf -d | 6.134 | 0.727 | 4.508 | 0.352 | 0.417 | 0.120 |
| Large *Ca. leptoporus* | Cal large | 6.370 | 0.931 | 3.737 | 1.053 | 1.336 | 0.092 |
| *H. carteri* | Hel | 8.936 | 0.994 | 2.541 | 1.740 | 2.440 | 0.032 |
| *Co. pelagicus* | Cpl | 10.640 | 1.175 | 10.610 | 0.950 | 1.235 | 0.094 |




**Figure 1**. Temporal and spatial distribution of samples. (a) The evolution of studied coccoliths: first

occurrence and last occurrence data are from Nannotax3

(http://www.mikrotax.org/Nannotax3/index.html). The blue bars represent ranges of first occurrence
and the green bars represent ranges of last occurrence. The blue diamonds represent samples used in
this study. (b) Spatial distribution of samples. 1304 means IODP U1304, 3428 means MD12-3428cq,
1433 and 1435 means IODP U1433 and U1435, respectively. 807 means ODP 807 and 21-2 means

KX21-2.

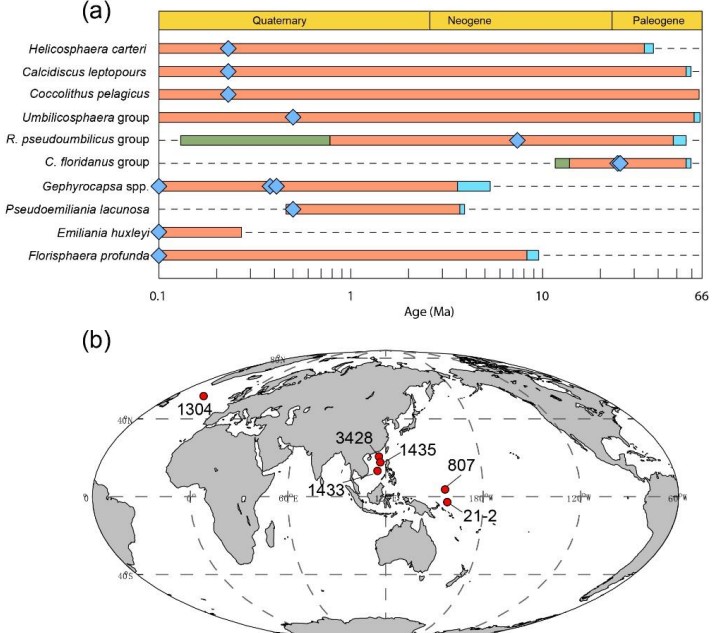







**Figure 2.** Schematic of settling experiments. The pictures were taken after *Coccolithus pelagicus*
sinking experiments with T=0 and T=30 min. V1 and V2 are the volumes of the upper and lower

cylinders, D is the settled distance.

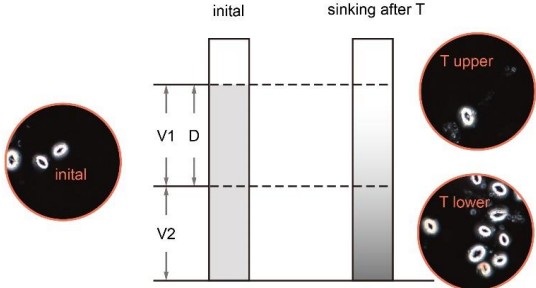





**Figure 3.** The shape parameters of vessels. V1 and V2 means the volume of upper suspension and
lower suspension, respectively. D means sinking distance. Φ means average inner diameter which is
calculated by $V1/(\pi D^2)$.

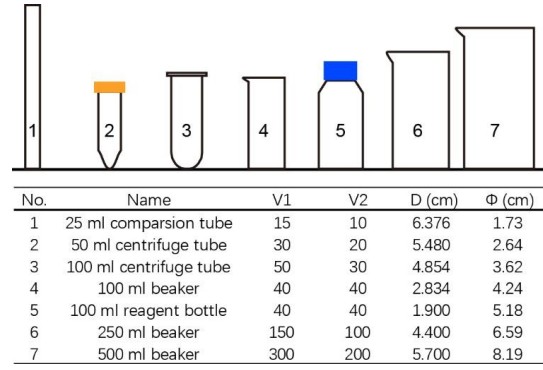

| No. | Name | V1 | V2 | D (cm) | Φ (cm) |
|---|---|---|---|---|---|
| 1 | 25 ml comparsion tube | 15 | 10 | 6.376 | 1.73 |
| 2 | 50 ml centrifuge tube | 30 | 20 | 5.480 | 2.64 |
| 3 | 100 ml centrifuge tube | 50 | 30 | 4.854 | 3.62 |
| 4 | 100 ml beaker | 40 | 40 | 2.834 | 4.24 |
| 5 | 100 ml reagent bottle | 40 | 40 | 1.900 | 5.18 |
| 6 | 250 ml beaker | 150 | 100 | 4.400 | 6.59 |
| 7 | 500 ml beaker | 300 | 200 | 5.700 | 8.19 |




**Figure 4.** Sinking velocities of *G. oceanica* in the core KX-21-2 measured in different vessels. (a) The
calibrated separation ratios measured in different vessels. Error bars show 95% confidence level of
calibrated separation ratio. (b-d) The relationship between sinking velocity and different vessel shape
parameters. Error bars represent 95% confidence level of sinking velocity in each vessel and the shade
area represents 95% confidence level of sinking velocity considering all data points.

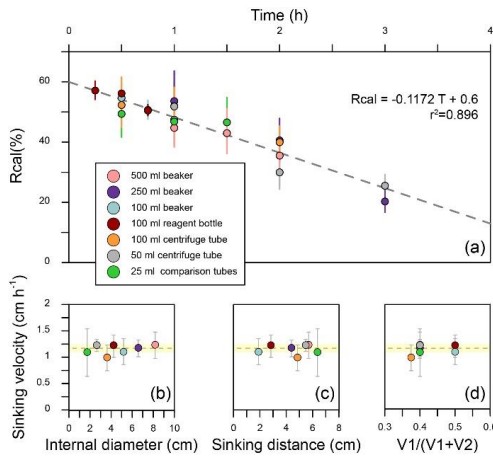




**Figure 5.** The calculated separation ratio (Rcal) vs sinking duration. Fp-WP means *F. profunda* in the
West Pacific. Fp-SCS means *F. profunda* in the South China Sea. Emi means *E. huxleyi*. Gspp means
small *Geophyocapsa*. Geo means *G. oceanica*. Gcarb means *G. caribbeanica*. Ret<4 means small
*Reticulofenestra*. Ret>4 means large *Reticuloenestra*. Cyf means *Cyclicargolithus floridanus*. Cy-d
means dissolved *Cy. floridanus*. Umb means *U. sibogae*. Pla means *Pseudomiliania lacunose*. Hel
means *Helicosphaera carteri*. Cal large means larger *Calicidiscus leptoporus*. Cal small means small

*Ca. leptoporus*. Cpl means *Co. pelagicus*.

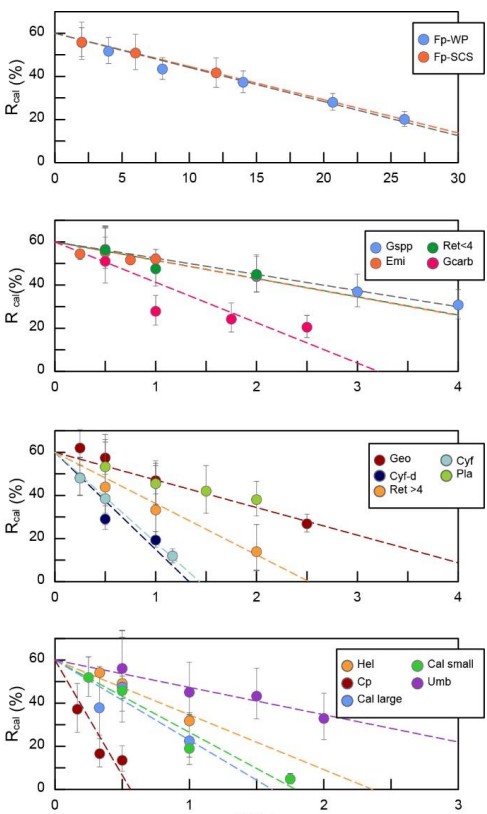







**Figure 6.** Coccolith sinking velocities and coccolith shape factors. (a-b) Sinking velocities and mean

distal shield length. The horizontal error bars represent one standard deviation of coccolith length and

the vertical ones represent 95% confidence level of measured sinking velocities. The blue, green and

red lines represent sinking velocity of calcite sphere objects, coccolith sinking velocities estimated by

Bolton et al. (2012) and this study, respectively. (c) The ratio of measured speed and speed calculated

by Stokes' Law. (d) Coccolith short axis length (SAL) and long axis length (LAL) ratio against shape-

velocity factor $k_{sv}$. Box shows median value and upper/lower quartiles, whiskers show maximum and

minimum values, outliers larger than 1.5 of the interquartile range are shown as red crosses. The SAL

against LAL plot was shown in Figure C3. The short names of coccoliths can be found in Table 2.

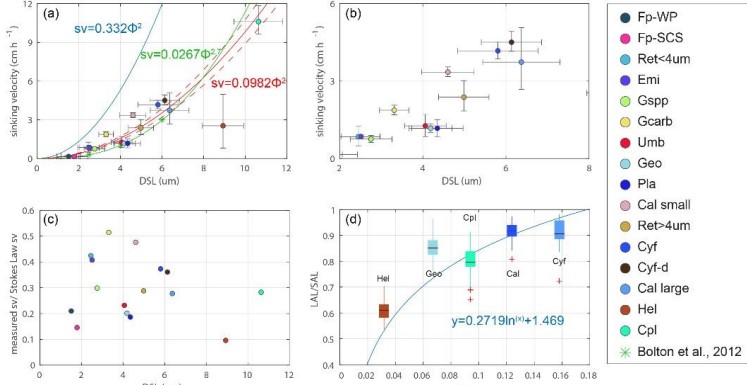




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





## Appendix A. Sample selections

**Table A1**. Sample selections

| Measured coccolith | abb. | Region | Core | Section | Epoch | Age model ref. |
|---|---|---|---|---|---|---|
| *F. profunda* | Fp-SCS | SCS | MD12-3428 | 0-1 cm | Holocene | Zhang et al., 2016 |
| *F. profunda* | Fp-WP | W.P. | KX21-2 | 2-4 cm | Holocene | Liang et al., 2016 |
| *E. huxleyi* | Emi | SCS | MD12-3428 | 0-1 cm | Holocene | Zhang et al., 2016 |
| *Gephyocapsa* spp. | Gspp | W.P. | ODP 807A | 1H 5W 102-104 | Pleistocene | Jin et al., 2010 |
| *G. oceanica* | Geo | W.P. | KX21-2 | 2-4 cm | Holocene | Liang et al., 2016 |
| *G. caribbeanica* | Gcarb | N.A. | IODP 1304B | 7H 5W 69-70 | Pleistocene | Channell et al., 2010 |
| small *Reticulofenestra* | Ret<4 | SCS | IODP 1433B | 28R 2W 30-34 | Miocene | Li et al., 2013 |
| large *Reticulofenestra* | Ret>4 | SCS | IODP 1433B | 28R 2W 30-34 | Miocene | Li et al., 2013 |
| *Cyclicargolithus floridanus* | Cyf | SCS | IODP 1435A | 6R 3W 25-29 | Oligocene | Li et al., 2013 |
| *Cyclicargolithus floridanus* | Cyf-d | SCS | IODP 1435A | 8R 1W 27-31 | Oligocene | Li et al., 2013 |
| *Umbilicosphaera sibogae* | Umb | W.P. | ODP 807A | 3H 5W 92-94 | Pleistocene | Jin et al., 2010 |
| *Pseudoemiliania lacunosa* | Pla | W.P. | ODP 807A | 3H 5W 92-94 | Pleistocene | Jin et al., 2010 |
| *Helicosphaera carteri* | Hel | W.P. | ODP 807A | 3H 5W 92-94 | Pleistocene | Jin et al., 2010 |
| large *Calcidiscus leptoporus* | Cal large | W.P. | ODP 807A | 3H 5W 92-94 | Pleistocene | Jin et al., 2010 |
| small *Calcidiscus leptoporus* | Cal small | N.A. | IODP 1304B | 7H 5W 69-70 | Pleistocene | Channell et al., 2010 |
| *Coccolithus pelagicus* | Cpl | N.A. | IODP 1304B | 7H 5W 69-70 | Pleistocene | Channell et al., 2010 |

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



**Appendix B. Coccolith images under circular polarized light**

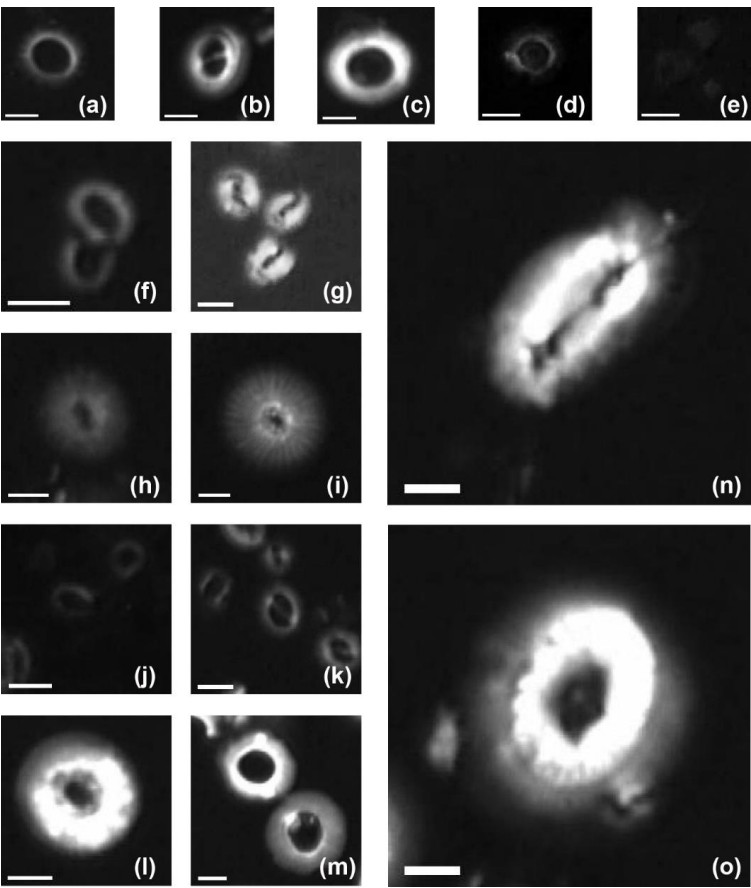

**Plate B1**. Imaged of measured coccolith in this study: (a) *Pseudoemiliania lacinosa* in the core ODP 807; (b) *Gephyrocapsa* oceanica in the core KX21-2; (c) *Reticulofenestra* spp. (large) in the core IODP U1433B; (d) *Umbilicosphaera sibogae* in the core ODP 807; (e) *Florispharea profunda* in the core KX21-2; (f) *Reticulofenestra* spp. (small) in the core IODP U1433B; (g) *Gephyrocapsa caribbeanica* in the core IODP U1304B; (h) small *Calcidiscus leptopours* in the core IODP U1304B; (i) large *Calcidiscus leptopours* in the core ODP 807A; (j) *Emiliania huxleyi* in the surface sediment in the South China Sea; (k) *Gephyrocapsa* spp. in the core ODP 807; (l) *Cyclicargolithus floridanus* in the core IODP U1435A and (m) dissolved *Cyclicargolithus floridanus* in the same core; (n) *Helicosphaera carteri* in the core ODP 807A; (o) *Coccolithus pelagicus* in the core IODP U1304B. White bars represent a length of 2 μm.





### Appendix C. The length distribution of coccoliths

To measure the distal shield length of coccoliths, pictures were taken at a magnification of 1250x

under circular polarized light. The coccolith lengths were measured by using the image analysis

software, ImageJ. More than 5 pictures were taken and more than 50 (usually more than 100)

coccolith specimens were measured. The length distributions of coccoliths measured in our

experiments were shown in the Figure C1.

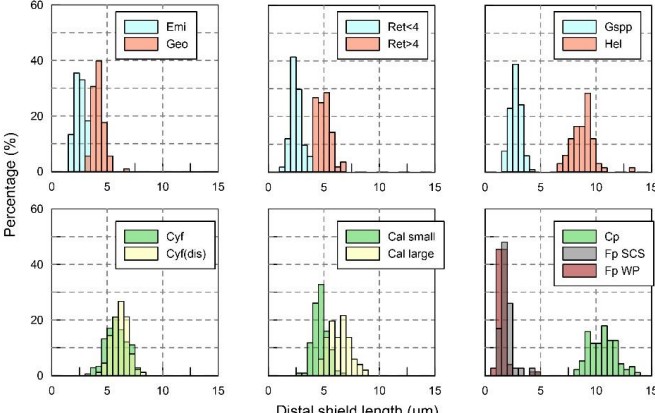

**Figure C1**. Size distribution of coccolith measured in the present study. The shorten names of coccolith

follow Table A1.

The classification of coccoliths by length was supported by mixture analysis in PAST (Hammer et

al., 2001), such as *Reticulofenestra* spp. and *Gephyrocapsa* spp. *Reticulofenestra* spp. in the

Miocene were classified into two groups, Ret. (<4 μm) and Ret. (>4 μm). The traditional

classification of *Reticulofenestra* spp. is <3 μm, 3-5 μm and 5-7 μm didn't pass the normal

distribution test. Hence, in this study the *Reticulofenestra* spp. are divided at 4 μm (Figure C2).

*Gephyrocapsa* spp. were classified by the shape of coccoliths into small *Gephyrocapsa* (central area

opening and length <3.5 μm), *G. oceanica* (central area opening and length >3.5μm) and *G.

caribbeanica* (closed central area) by the length and central area.



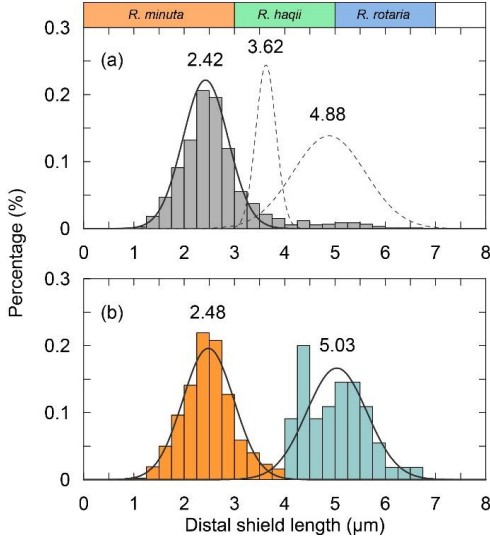

**Figure C2**. The classical classification of *Reticulofenestra* spp. (a) and the classification used in our

study (b). The curves represent the normal distribution fits of different coccolith groups and the dish

curve marks that the goodness of fit is below 0.2.

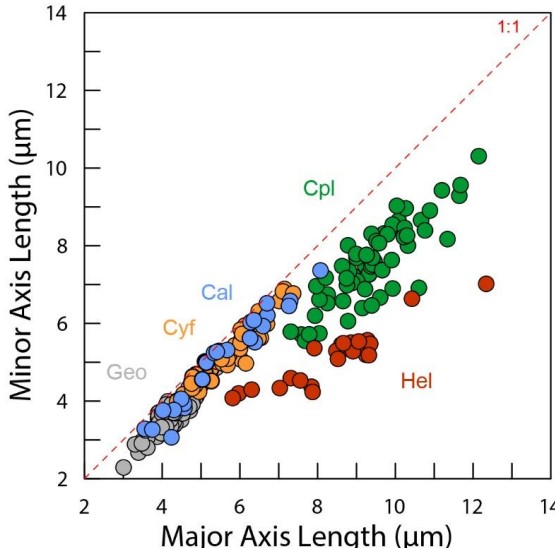

**Figure C3.** The short axis and long axis length distribution of coccoliths in Figure 6d.

**Reference.**



Hammer, Ø., Harper, D., Ryan, P., 2001. Paleontological Statistics Software: Package for

Education and Data Analysis. Palaeontologia Electronica.





### Appendix D. Coccolith movement in gravity settling

In this part, the derivation of equation will be explained in detail including proofs of several

assumptions mentioned in the methods part.

When the well mixed sediment begins to sink, the decrease of coccoliths number in the upper

suspension ($N_u$) can be described as following equation:

$$\frac{dN_u}{dT} = -\frac{N_{u(t=0)}}{D} \times sv \qquad \text{(D-1)}$$

where the D is the length of upper suspension and $N_{u(t=0)}$ /D is the initial number of coccolith in

cross-section with a thickness of dD.

Do integration for the equation D-1, we can get the variation of coccolith number in the upper

column over time:

$$N_u = N_{u(t=0)} - \frac{N_{u(t=0)}}{D} \times sv \times T \qquad \text{(D-2)}$$

After a period of time (T), we pump out the upper suspension. Here we define the number of

coccoliths in the upper supernatant dividing the total coccoliths number in the tube (Nt) as separation

ratio (R), which represents the percentage of total coccoliths removed in one separation. This

parameter is important and will be repeatedly mentioned in the following part. R can be expressed

by

$$R = \frac{N_u}{N_t} \qquad \text{(D-3)}$$

Assuming all coccoliths are uniformly distributed in the suspension at the beginning of settling,

Nu(t=0) has relationship with Nt as follow:

$$\frac{N_{u(t=0)}}{N_t} = \frac{V1}{V1+V2} \qquad \text{(D-4)}$$

where V1 is the volume of upper suspensions and V2 is the volume of lower suspensions.

Combining the equation D-1, D-2, D-3 and D-4, we obtain the relationship between separation ratio,

R, and sinking velocity, sv, as follow:

$$R = \frac{N_u}{N_t} = \frac{N_{u(t=0)} - \frac{N_{u(t=0)}}{D} \times sv \times T}{N_t} = \frac{V_1 - \frac{V_1}{D} \times sv \times T}{V_1 + V_2} \qquad \text{(D-5)}$$

If we plot the R and T on a figure, the slope of the line is a function of V1, V2, D and sv. Since the

V1, V2, D are known parameters, we say the slope of R-T is a function of sv, which is exactly what

we want.





Comparison tubes used in our experiments have the same V1 and V2 but different D. Other vessels
used in other experiments have different V1, V2 and D. So we should adjust the raw separation ratio
to calibrated separation ratio (Rcal), which represents the separation ratio made in a standard vessel
with V1=15 ml, V2=10 ml and D=6 cm. This step can be described by equation 2-6:
$$Rcal = \frac{[R \times (V1+V2) - V1] \times D \times 15}{(6 \times V1 + 15) \times 25}$$    (D-6)
After calibrated, the slope of Rcal-T (k) has relationship with sv as following equation:
$$sv = -10 \times k$$    (D-7)
Hence, the sinking velocity of different coccolith can be achieved by measuring the variations of
Rcal over time.
We also offer a test for the assumption that the average sinking velocity of all coccoliths can be
treated as the sinking velocity of coccoliths with the average length. Here we used the data of G.
oceanica. A normal distribution was fitted to the measured length distribution (Figure D1-a). And
then we simulate a normal distribution situation of coccoliths in the vessel. The sinking velocities
of different size coccoliths were calculated by the cubic shape parameter 'b' as described in
discussion part. We modeled the coccoliths sinking process and computed the separation ratio
(Figure D1-b), coccolith length (Figure D1-c) and instant sinking velocities (Figure D1-d) at
different time sections.

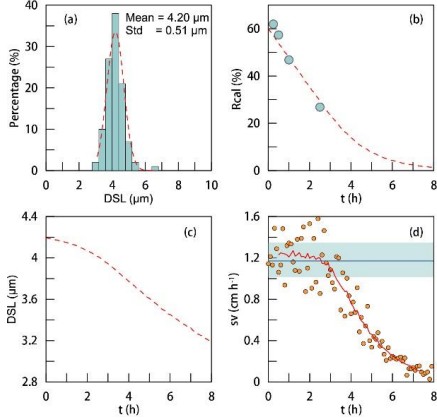




**Figure D1**. The simulations of coccoliths settling with different lengths: (a) the length distribution of
coccoliths. The green bars represent measured data and red dash line represents the best fit for normal

distribution. (b) The calibrated separation ratio: the green dots are measured data in our settling

experiments and the red dash line represents results obtained from Monte Carlo simulations. (c) The
average length of removed coccolith in simulations; (d) the sinking velocities of coccoliths: the orange

dots are instant sinking velocity calculated from derivation of Rcal, the red dash line is weighted

average for the instant sinking velocity. Blue line represents the average sinking velocity we measured

and the green shade area represents 95% confidence level of the measured velocity.

For *G. oceanica* experiments, the instant sinking velocity would not change significantly until
settling for more 3 hours. That means for all Rcal larger than 15% are safe for liner regressions. The
minimum safe number of Rcal will descend with the drop of dispersion degree of coccolith length
distribution. Hence our assumption for average sinking velocity and the use of liner regression are
proved to be reasonable.



## Appendix E. Statistical and error analyses

The errors of measured separation ratio (R) and calculated sinking velocity (sv) are mainly caused

by counting coccolith, the error of which fellows the Poisson distribution. To detect the influence of

counting number on the result error, the error of separation ratio was simulated by 5000 times Monte

Carlo calculations with assumptions that 'V1:V2=15:10' and 'n1=n2' (Figure E1). The result shows

that the number of coccolith counted in the upper column draws more influence on the relative error

($|R-R_{95CL}|/R$). That means more coccolith in the upper suspension should be counted to make results

more accurate. The slope of R-T was calculated by liner fitting with the intercept fixed on

V1/(V1+V2). The error of sinking velocity was also calculated by 5000 times Monte Carlo

simulations in the software Matlab.

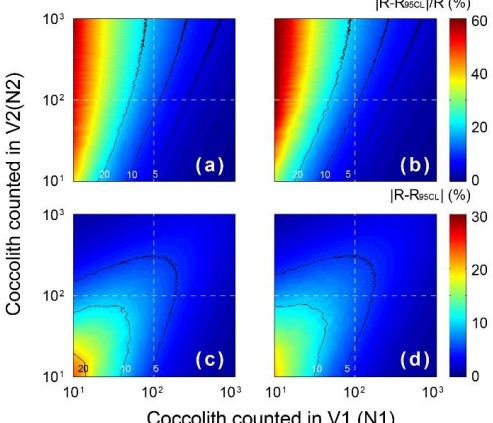

**Figure E1**. The error distribution with different N1 and N2 (ranging from 1 to 1000) simulated 5000

times by the Matlab with assumptions that the error distributions of N1 and N2 fellow Poisson

distribution. The calculation of R follows equation 2-5, and here we assume numbers of FOV are equal

(n1=n2). Counter lines mark values equal to 5, 10 and 20. (a) and (c) represent the lower 95%

confidence level and (b) and (d) represent upper 95% confidence level. (a) and (b) the relative error of

R and (c) and (d) represent the absolute error of R.