# Peer review of "A refinement of coccolith separation methods: Measuring the sinking"

_Biogeosciences, 2018_

## Referee Comment (RC1) · Anonymous Referee #1 · 18 Mar 2018

Dear Editor,

I have reviewed the Technical Note entitled 'A refinement of coccolith separation methods: Measuring the sinking characters of coccoliths' submitted by Zhang et al. to Biogeosciences.

In this study, the Authors have measured the settling velocity of a selection of sedimentary coccoliths and claim that their new dataset – and the various parameters linking size/shape and the speed of decanting by gravity in aqueous solution– will be of use for the microseparation protocol of these calcareous nannofossils. Indeed, these micron-sized calcite particles are impossible to isolate under the binocular microscope, as done in routine for the foraminifera. Yet, recent works highlight the potential of the biominerals produced by the coccolithophores in palaeoceanographic research. I am

generally supportive of publication of this work in Biogeosciences – but I have a number of comments / questions that should be addressed in a future version of the draft, would the Authors decide to follow my points. I am not really familiar with the decanting technique for concentrating coccoliths, but my overall impression is that final users will find hard to use the data of this paper to facilitate/enhance the processing of their own samples.

- In my opinion, the lack of the integration of the coccoliths with coincident particles (quartz, clays, other calcite particles, including other coccoliths) represents a major caveat of the refinement of the decanting protocol. The Authors treat their assemblages as monospecific coccolith assemblages. For the large assemblages, which yielded 50% relative abundance of the target species, what is the effect of other calcite particles? If their composition change, would that change the settling velocity? More importantly, it is well known that clays are charged particles that are able to form aggregates ('flaks') in suspension and as such, these particles are prone to substantially influence the setting velocity. This issue is only briefly acknowledged by the 'hindered settling'. This is crucial for the application of the parameters in natural assemblages containing various concentrations (?nature) of clay minerals. Therefore I am of the opinion that this points need to be further discussed. Adding synthetic clay minerals in the assemblages would have been a sensitive means to address this criticism, although I am not advocating that the Authors should perform more experiments.

- It is not clear to me how many particles (coccoliths) were actually counted, nor if replicated measurements have been conducted? Also, it would be good to explain the 'drop technique' used in this study.

- It is not clear from reading the text why Helicosphaera carteri escapes the settling velocity equation derived for other taxa (L203-205).

- Why is the potential of centrifuging not discussed at all - except a brief mention L47?

- Figure 1 should include the array of sizes of the various coccoliths presented.

- Figure 2 is not really convincing given the number of coccoliths in the field of view.

Minor comments:

L87: That Pseudoemiliania lacunosa and Umbilicosphaera sibogae are impossible to differentiate is premature here, and should be discussed later in the manuscript.

L143 "in ammonia at 20°C" – I guess you mean in deionized water neutralized by addition of ammonia? L348 : Publication date is 2009. L415 Pseudoemiliania lacunosa is mispelt. L420 Calcidiscus leptoporus is mispelt. (Many other taxa are misspelt throughout the text and captions).

―――――――――――――――――

---

## Referee Comment (RC2) · Anonymous Referee #2 · 27 Mar 2018

In this manuscript Zhang et al. investigate empirically the settling rates of sedimentary coccoliths of different sizes and shapes in the laboratory. Repeated settling is a technique that is widely used to separate natural sediment assemblages into near mono-specific fractions, but as far as I am aware this is the first attempt to calibrate this approach quantitatively over a range of species. The dataset presented by Zhang et al. will be a useful contribution to the sedimentary coccolith literature, and I believe Biogeosciences is a good place to present this work. However I do have a number of comments.

In general the manuscript is well-written, although there are parts that feel muddled and are difficult to understand - especially where equations are derived. The motivation for the research is not particularly well laid out (why do we care about obtaining mono-

specific fractions?). The dataset is good, but is not presented in a particularly helpful way for anyone who wants to use their results.

The importance of this work is in the practical laboratory application of separating coccoliths, rather an advance in scientific understanding. In this manuscript in its current form, this is lost behind the emphasis of settling velocities. The importance of settling velocities of individual suspended coccoliths in the lab is hard to appreciate; this isn't a quantity that can be used directly for the purpose of species separation in the lab, and its biogeochemical importance in the natural environment is questionable. I would suggest that the authors consider reframing this work as a tool for subsequent authors to separate mixed coccolith samples into monotaxonomic fractions: Specifically to describe their suggested protocol, and how to calculate ideal combination of settling times and (e.g. as a trade off between yield and quality of separation?). If they do wish to comment on a comparison between settling rates observed in the laboratory, and those observed in sediment traps, I think this needs at least a full paragraph in the discussion.

The assumptions underpinning this work should be more clearly stated in the main text. Firstly, all coccoliths belong to a particular species are assumed to sink at exactly the same rate. Secondly, they are assumed to sink at a constant velocity from the instant that the suspension is left. I would like to see a calculation in the appendix estimating the time and distance that a particle falls before it reaches terminal sinking velocity, to show whether or not it is justifiable to ignore the accelerating phase for all of the particle sizes considered here. Intuitively I imagine this is a fair assumption, but it would be nice to see in numbers. Furthermore, I think it would be useful for the authors to test their approach with spheres (such as spherical glass beads) of similar size to coccoliths such as those used to calibrate Coulter counters. The authors have made theoretical calculations based on idealized spheres (and compared the differences between the observed values in lines 211-212), so this sort of approach would be a good test of their proposed protocol - elucidating the degree to which differences between an idealized

scenario in theory and observations of coccoliths is due to shape, the experimental set up, or assumptions made in the calculation of their parameter, R. They could also spike a sample of sedimentary coccoliths with these beads in order to test the density of suspension that leads to hindered settling.

The authors justify the assumption that settling rates are approximately constant with a time course analysis of *Gephyrocapsa oceanica*, concluding that for the first 4 hours, settling velocities do indeed appear to be constant. Is this period of 4 hours applicable across coccoliths of other size and shape? What causes the deviation from the ideal stokes law behaviour after 4 hours? If this were an ideal scenario, the top part of the vessel should be completely devoid of coccoiliths of a given size after a period of time T, where $T = \frac{D}{sv}$. I would like a more in depth discussion of these features and other factors affecting sinking velocities in the lab - for example - temperature gradients leading to convection, entrainment of small particles by larger ones (i.e. do smaller coccoliths sink faster when there are large coccoliths present?).

8-9 I suggest that the authors remove the reference to CaCO3 export from the surface ocean. In the ocean, sinking velocities are greatly complicated by flocculation with organic matter, and through grazing - as mentioned in line $\sim$ 178, most coccoliths probably ended up in sediment packaged up in larger aggregates such as faecal pellets.
It would be useful however to have the complexities of the real ocean alluded to much more clearly and earlier in the manuscript, so that readers are not tempted to use these calculations to estimate export rates directly from individual coccoliths in sediment.

24-38 From a non-specialist point of view it is not clear from the first paragraph why it is desirable to obtain monospecific fractions.

Eq. 2-2 test this equation in an ideal scenario using glass spheres?

150 This doesn't make sense

89-119 I think this section would benefit from being slightly more thorough and clear about how the proposed protocol is actually implemented. For example: I assume that when counting coccoliths in the lower part of the settling vessel, that the remaining suspension must be homogenized, including re-suspending any coccoliths that have settled out, before counting. If so, this should be stated explicitly.

162-164 "sediments accumulating in the lower suspension, the particle concentration can be more than 4 times higher than the initial homogenous concentration" – This is important and should be discussed thoroughly. How do these higher concentrations arise? Presumably due to the size range of coccoliths in the sample. Can this effect be described quantitatively as a function of the standard deviation of coccoliths sizes in the initial sample?

179 "confirming the fact" is far too strong. It is true that these numbers are consistent.

189 Why is H. carteri excluded?

Table. 2 I assume that the asymmetrical uncertainties on sinking velocity may arise due to an assumed normal distribution of coccolith size via the quadratic relationship? If so, this should be stated.

Fig.2 This figure doesn't really represent the assumptions made by the authors. For coccoliths of a given size, the boundary between the suspension and the supernatant is infinitely sharp, and the suspension does not change in density - but rather there is a build up of coccoliths deposited on the bottom of the vessel. In a mixed species assemblage, or where coccoliths are a range of sizes, then the suspension will become more dense towards the bottom over time as shown here, but this isn't currently represented in the equations (or at least not clearly!).

For this reason, these coccolith images are fairly unhelpful. A schematic figure that more clearly shows the change in coccolith density might be better, with a more obvious range in sizes (or not).

Fig.2 If the authors are using the volume and sinking distance to estimate the average vessel diameter, the equation given in the caption doesn't look right. I think it should be: $\phi = 2 \times \sqrt{\frac{V_1}{\pi D}}$.

Appendix D While the math seems sensible, I found it difficult to follow this derivation despite its simplicity. Nevertheless, the way of measuring sinking velocity proposed here is interesting, and I would personally prefer to see its derivation in the main text rather than the appendix. Specific points:

- Each variable should be defined after it is first used throughout the text, and again within the appendix if this is to constitute a stand alone derivation.
- A single symbol would be better for sinking velocity unless either 's' or 'v' is subscripted.
- If sv is a function of t, show this. If not, and you're interested in the average sv, I think
- The ratio given in line 458 is not the number of coccoliths in a thickness dD as stated - as the authors have defined here, it is the number of coccoliths per unit unit thickness.
- Figure D1: What does Monte Carlo mean in b) here? Have the parameters of the model been fitted to the data points multiple times, resampling their values from an assumed distribution? If so, the spread of constrained these values rather than just the average needs to be plotted to show how uncertain this relationship is. I assume that the early, straight part of the line in b) is the part that is described by equation 2-2, before the settling velocities

decrease when the suspension is left for 4 hours (d) - if so, it would be helpful to plot this straight line on here too and label it as the fit to equation 2-2 in the valid region. I don't understand how the authors obtain the shape of the relationship in b), so would benefit from further explanation. Why are there more data points in d) than in b)?

482 equation 2-6 doesn't exist. Should this be D-6?

eq. D-6 This is difficult to follow. Keep equation in symbol format before introducing numbers.

eq. D-7 What is -10, and what is k?

Appendix E It's not clear to me how a Monte Carlo approach has been used here, nor the benefits of using such an approach over propagation of uncertainty equations. As far I understand it, the authors have simply calculated the uncertainty associated with equation 2-1, for a range of explicit values of N1 and N2.

---

## Referee Comment (RC3) · Anonymous Referee #3 · 31 Mar 2018

Dear Editor,

I have reviewed the manuscript entitled "A refinement of coccolith separation methods: Measuring the sinking characters of coccoliths" by Zhang et al. The repeat settling/decanting method of separating coccoliths from natural sediment is widely used in routine laboratory analyses. To improve this method is very important to promote the nannofossil based geochemical analyses and paleoceanographic research. The authors have selected a good point to calibrate the coccolith sinking velocity and discussed the different influencing factors. The experimental design is reasonable, the dataset and results are elaborative and informative. In my opinion, this manuscript is worth publishing in such high ranking journal as Biogeosciences. However, this paper still needs some revisions before it could be accepted for publication. My comments

are below:

1. The authors took most of the paragraphs to describe and calculate the sinking velocities of different coccolith species. However, what is the application of this parameter in future research? This is not very clear to me. I think the purpose of this paper is to give the audience "a refinement of coccolith separation method". So I suggest adding some paragraph to introduce how to use your SV data in routine work or to give the audience some suggestions how to improve the efficiency or precision of the separation method after your work.

2. In this manuscript, the authors used several technique and methods in the experiments, such as "sinking method or filtering method" in L83 or "drop technique" in L99. This would be difficult to follow for the audience who are not very familiar with coccolith separation. I suggest adding some brief explanations of these techniques.

3. The authors selected eight raw sediment samples from different cores in global oceans. As I know, these cores have different geographic settings like different water depths, mineral composition and nannofossil preservation. Do these factors influence the separation process or the sinking velocity?

4. The section of "Conclusions", this part is more or less like a part of discussion and not so constructive to me. I suggest improving this part.

5. In L86-87, "except the Pseudoemiliania lacunosa and Umbilicosphaera sibogae, which cannot be separated from each other". Why? Should give some explanations.

6. In L188-189, "If we use data for all species except Helicosphaera carteri..." why don't include H. carteri in the calibration?

7. L66, change "two Neogene samples" to "two Neogene/Paleogene samples"

---

## Author Comment (AC1) · 21 Apr 2018

Thank you for your thoughtful suggestions and comments.

**Comments 1:**
In my opinion, the lack of the integration of the coccoliths with coincident particles (quartz, clays, other calcite particles, including other coccoliths) represents a major caveat of the refinement of the decanting protocol. The Authors treat their assemblages as monospecific coccolith assemblages. For the large assemblages, which yielded 50% relative abundance of the target species, what is the effect of other calcite particles? If their composition change, would that change the settling velocity?
**Reply:** Good question. If our understanding is correct, we think your question can also be asked in another way: what's the behavior of particles in a multi-species or multi-particle type settling system?
Based on the work by Masliyah (1979), we know that when particle concentration is smaller than 10%, the collisions among particles don't have significant influence on sinking velocity. Hence, in a situation where there are several kinds of particles in the suspension, if the concentration is low enough, we can treat them as independence settlings. In our experiments, because the particles concentration were below 5%, we therefore think that varying composition of the suspension would have a negligible influence on the measured sinking velocities. This would be the case for a multi-species coccolith assemblage or one with different particle times, so long as total particle concentration remained low.

**Comments 2:**
More importantly, it is well known that clays are charged particles that are able to form aggregates ('flaks') in suspension and as such, these particles are prone to substantially influence the setting velocity. This issue is only briefly acknowledged by the 'hindered settling'. This is crucial for the application of the parameters in natural assemblages containing various concentrations (?nature) of clay minerals. Therefore I am of the opinion that this points need to be further discussed. Adding synthetic clay minerals in the assemblages would have been a sensitive means to address this criticism, although I am not advocating that the Authors should perform more experiments.
**Reply:** We emphasize that 'hindered settling ' is different to 'settling as aggregates'. As we mentioned in Lines 154-158 (Lines 134-141 in the former version), 'hindered settling' was caused by high concentration of suspension and collisions among particles. In the sample of ODP 807, there are aggregates in raw sediments even after 24 hours soaking in 0.2% ammonia. There is a protocol using benzalkonium chloride to disaggregate (Minoletti et al., 2008). In our pretreatments for this site, we discarded any large rapidly sinking aggregates that remained after soaking, before proceeding with settling steps. We now mention this in Lines 91-92. We acknowledge that, in this study, we have not tested the direct effect of disaggregated clays present in the solution on coccolith settling rates. However, as mentioned above, low concentrations should minimize these effects.

**Comments 3:**
It is not clear to me how many particles (coccoliths) were actually counted, nor if replicated measurements have been conducted? Also, it would be good to explain the 'drop technique' used in this study.
**Reply:** Thank you for pointing this out. In most experiments, more than 300 (usually around 500) coccoliths were counted. For *H. carteri*, we counted more than 100 FOVs and about 100 specimens because the number of *H. carteri* is much smaller than other coccolith even after pretreatments. We have added this statement in the new version (Lines 113-119).

**Comments 4:**
It is not clear from reading the text why Helicosphaera carteri escapes the settling velocity equation derived for other taxa (L203-205).
**Reply:** Thanks for pointing out that this needs clarification. We didn't use *H. carteri* in the regression because of its specific shape, which is quite different to the other species studied. This was explained it in Lines 224-233 (Lines 203-205 in the former version): the ellipticity of *H. carteri* (~0.6) is significant lower than other coccolith (among 0.8-0.9), therefore its settling behavior differs from other species. This is also illustrated in Figure 6d and Figure C3. We have reorganized this part to make it clearer.

**Comments 5:**
Why is the potential of centrifuging not discussed at all - except a brief mention L47?
**Reply:** This is a good point and a question that we are currently working on. To data, we have calculated the movement of coccoliths in a centrifuge machine and tried to use centrifuging instead of gravity settling. The centrifuging method works well for small coccolith such as *F. profunda* and *E. huxleyi*. However, the uncertainty will become larger when we try to separate large coccolith such as *C. pelagicus*. We are still working on improving the centrifuging method, which would be the subject of a future publication and we would prefer to focus on sinking velocity measurements under gravity in this study.

**Comments 6:**
Figure 1 should include the array of sizes of the various coccoliths presented.
**Reply:** Thank you for your suggestion. We have listed mean sizes and standard deviations of size in Table 2. Since Figure 1 showed the evolutionary ranges timing of different coccolithophores and the coccoliths' size for each species varied in geological time, we think plotting the size data in our sample with fixed values on Figure 1 could be misleading.

**Comments 7:**
Figure 2 is not really convincing given the number of coccoliths in the field of view.
**Reply:** We have redrawn Figure 2 and replaced these photos by a schematic drawing. See the new version Figure 2.

**Comments 8:**
That Pseudoemiliania lacunosa and Umbilicosphaera sibogae are impossible to differentiate is premature here, and should be discussed later in the manuscript.
**Reply:** We have changed this sentence as '*Pseudoemiliania lacunosa* and *Umbilicosphaera sibogae* were measured together' and we have modified the former 'Conclusions' part as 'Suggestions for coccolith settling velocity estimations and separations' and you can find an explanation in Lines 277-279.

**Comments 9:**
L143 "in ammonia at 20_C" – I guess you mean in deionized water neutralized by addition of ammonia?
**Reply:** Yes. We have changed this sentence as 'in 0.2% ammonia at 20℃'.

**Comments 10:**
L348 : Publication date is 2009. L415 Pseudoemiliania lacunose is mispelt. L420 Calcidiscus leptoporus is mispelt. (Many other taxa are misspelt throughout the text and captions).
**Reply:** We apologize for the spelling mistakes. We have done double checks in this new version. We have changed publication data in Line 34, 88 and 411.

**References:**
Masliyah, Jacob H. "Hindered settling in a multi-species particle system." Chemical Engineering Science 34.9 (1979): 1166-1168.

[revised manuscript text omitted]

---

## Author Comment (AC2) · 21 Apr 2018

We thank the reviewer for his/her thoughtful and constructive questions and suggestions on our manuscript, which will improve the clarity and the quality of the paper.

**Comments 1:**

8-9 I suggest that the authors remove the reference to CaCO3 export from the surface ocean. In the ocean, sinking velocities are greatly complicated by flocculation with organic matter, and through grazing - as mentioned in line 178, most coccoliths probably ended up in sediment packaged up in larger aggregates such as faecal pellets. It would be useful however to have the complexities of the real ocean alluded to much more clearly and earlier in the manuscript, so that readers are not tempted to use these calculations to estimate export rates directly from individual coccoliths in sediment.

**Reply:** Agreed. We have removed the reference as your suggestion.

**Comments 2:**

24-38 From a non-specialist point of view it is not clear from the first paragraph why it is desirable to obtain monospecific fractions.

**Reply:** Thank you, we have tried to clarify this. Published data show that coccoliths have strong species and/or size-species vital effects in oxygen and carbon isotope and in elemental ratios (e.g. Ziveri et al., 2003; Rickaby et al., 2007; Stoll et al., 2012; Hermoso et al., 2016; Mejia et al., 2018). To be able to glean useful information from the geochemistry of fossil (or water sample) coccoliths, it is therefore desirable to try to separate monospecific or size-restricted fractions, which will provide more precise information on the past environment than a mixed coccolith fraction. We have added this in the new manuscript version.

**Comments 3:**

Eq. 2-2 test this equation in an ideal scenario using glass spheres?

**Reply:** Thank you for your suggestion. However in the context of our study, we think the principle of the (theoretically-derived) equation is clear and it is not necessary to design a new experiment to prove it.

**Comments 4:**

This doesn't make sense

**Reply:** We are not sure of the source of confusion. Please clarify this comment.

**Comments 5:**

89-119 I think this section would benefit from being slightly more thorough and clear about how the proposed protocol is actually implemented. For example: I assume that when counting coccoliths in the lower part of the settling vessel, that the remaining suspension must be homogenized, including re-suspending any coccoliths that have settled out, before counting. If so, this should be stated explicitly.

**Reply:** Constructive suggestion. Your guess is correct and we have added some more descriptions on this measurement (Lines 101-119).

**Comments 6:**

a) 162-164 "sediments accumulating in the lower suspension, the particle concentration can be more than 4 times higher than the initial homogenous concentration" – This is important and should be discussed thoroughly. How do these higher concentrations arise? Presumably due to the size range of coccoliths in the sample. Can this effect be described quantitatively as a function of the standard deviation of coccoliths sizes in the initial sample?

b) Figure 2 This figure doesn't really represent the assumptions made by the authors. For coccoliths of a given size, the boundary between the suspension and the supernatant is infinitely sharp, and the suspension does not change in density – but rather there is a build up of coccoliths deposited on the bottom of the vessel. In a mixed species assemblage, or where coccoliths are a range of sizes, then the suspension will become more dense towards the bottom over time as shown here, but this isn't currently represented in the equations (or at least not clearly!). For this reason, these coccolith images are fairly unhelpful. A schematic figure that more clearly shows the change in coccolith density might be better, with a more obvious range in sizes (or not).

**Reply:** These two comments are talking about the same issue: "Will the coccolith concentration be higher in the lower suspension during the settling?" and "if so, what caused this phenomenon?" We can share our experience and try to explain these to you.

We don't think the variations of coccolith shape can cause a significant increase of sediment concentration in the lower suspension (it can, but it is not significant). This is because we have pre-separated coccoliths from sediment before measurement and the coccoliths were not in a wide size range. The concentration of suspension really increased in some situations and could be seen with naked eyes. This often happened when we used centrifuge tubes. We observed the sediment concentration increased at the depth where the shape of vessel narrowed. So we think this phenomena was caused by the friction of the vessel wall and collision between particles. Precisely calculation this process is too complex and beyond the scope of paper. Importantly, because we only pump out the upper suspension in each vessels, the raise of concentration around bottom has not affected our result. We have added a new sentence in Lines 185-186 to avoid misunderstanding.

We made a mistake in original Figure 2, in which the sediment concentration variation had been overstated. We have redrawn this figure to correct this. We sincerely appreciate your carefully reviewing.

**Comments 7:**

"confirming the fact" is far too strong. It is true that these numbers are consistent

**Reply:** We have changed confirming to suggesting (Line 202).

**Comments 8:**

Why is H. carteri excluded?

**Reply:** Thanks for pointing out that this needs clarification. We didn't use *H. carteri* in the regression because of its specific shape, which is quite different to the other species studied. This was explained it in Lines 224-233 (Lines 203-205 in the former version): the ellipticity of *H. carteri* (~0.6) is significant lower than other coccolith (among 0.8-0.9), therefore its settling behavior differs from other species. This is also illustrated in Figure 6d and Figure C3. We have reorganized this part to make it clearer.

**Comments 9:**

a) I assume that the asymmetrical uncertainties on sinking velocity may arise due to an assumed normal distribution of coccolith size via the quadratic relationship? If so, this should be stated.

b) Appendix E It's not clear to me how a Monte Carlo approach has been used here, nor the benefits of using such an approach over propagation of uncertainty equations. As far I understand it, the authors have simply calculated the uncertainty associated with equation 2-1, for a range of explicit values of N1 and N2.

**Reply:** These comments are about the error estimation and we reply to them together.

We suggested that difference in uncertainties was caused by the error of $R_{cal}$ (Figure 5) and coccolith shape distributions were never involved in the sinking velocity calculation. In Figure 5, the positive direction error bars are often larger than negative ones and we think this was caused by the Poisson distribution of uncertainty in coccolith counting. So when we do the regression (this regression was also a Monte Carlo process), we will find the uncertainty of slope (sinking velocity=-10*slope) is asymmetric. That is the source of asymmetrical uncertainties.

The Monte Carlo method is a common method for error propagation and is suitable for our study for three reasons. Firstly, no matter how complex the target equation is, what we need to do is choose the right error distributions for some independent variables, by running the code and collecting the results. This can save a lot of time compared with partial differential equation derivation.

Secondly, traditional error propagation assumpts that all uncertainties have a normal distribution. However, as we describe in Appendix E, the error distribution of coccolith counting is a Poisson distribution. Although when the number is large enough the Poisson distribution can be treated as normal distribution, in our study,  there were only around 10 coccoliths or even less in many FOVs and we input each FOV data independently. So we think the Monte Carlo method with exact error distribution is more suited to our data.

The last reason is that if we use the Monto Carlo method, we can take full advantage of uncertainty in the regression process. Otherwise, the liner regression will not consider the distribution  of uncertainties of the input data in a single regression and we will lose information related to coccolith counting errors. That is why we employed the Monte Carlo method for error propagation rather than using the partial differential equations.

In the revised version, we have reorganized the Appendix E to clarify how we did the Monte Carlo process (Line 587-591). Because the method is a common one, we don't think it is necessary to explain all of the above in the paper.

**Comments 10:**

If the authors are using the volume and sinking distance to estimate the average vessel diameter, the equation given in the caption doesn't look right. I think it should be:

**Reply:** Thank you for pointing out this mistake. We have checked the original equation in excel to make sure the calculation results are based on the correct formula.

**Comments 11:**

Appendix D: While the math seems sensible, I found it difficult to follow this derivation despite its simplicity. Nevertheless, the way of measuring sinking velocity proposed here is interesting, and I would personally prefer to see its derivation in the main text rather than the appendix.

**Reply:** Thank you for this suggestion. We have discussed this among co-authors. In previous versions, these derivations were indeed in the main text. We moved them to the appendix for a smoother reading experience. For those who wants to see details, they can check the appendix. So we want to keep them in the appendix and we have tried to make every equation clearer. If there are still some discontinuities in logic, please let us know.

**Comments 12:**

Each variable should be defined after it is first used throughout the text, and again within the appendix if this is to constitute a stand alone derivation. A single symbol would be better for sinking velocity unless either 's' or 'v' is subscripted.

**Reply:** Thank you for this suggestion. We have redefined the symbol, such as turning sv to v and V1 to $V_1$.

**Comments 13:**

   a) The authors justify the assumption that settling rates are approximately constant with a time course analysis of Gephyrocapsa oceanica, concluding that for the first 4 hours, settling velocities do indeed appear to be constant. Is this period of 4 hours applicable across coccoliths of other size and shape? What causes the deviation from the ideal stokes law behaviour after 4 hours? If this were an ideal scenario, the top part of the vessel should be completely devoid of coccoiliths of a given size after a period of time T, where T = D sv .
   b) If sv is a function of t, show this. If not, and you're interested in the average sv, I think
   c) Figure D1: What does Monte Carlo mean in b) here? Have the parameters of the model been fitted to the data points multiple times, resampling their values from an assumed distribution? If so, the spread of constrained these values rather than just the average needs to be plotted to show how uncertain this relationship is. I assume that the early, straight part of the line in b) is the part that is described by equation 2-2, before the settling velocities decrease when the suspension is left for 4 hours (d) - if so, it would be helpful to plot this straight line on here too and label it as the fit to equation 2-2 in the valid region. I don't understand how the authors obtain the shape of the relationship in b), so would benefit from further explanation. Why are there more data points in d) than in b)?

**Reply:** These questions concern the assumption "we treated the average sinking velocities as the sinking velocities of the coccoliths with the average length" in lines 138-140 and its proof in Appendix D.

Actually, the average sinking velocity is a function of t and that is why the modeled $R_{cal}$ and instant sinking velocity deviated from the ideal stokes law behaviour after 4 hours. The fundamental reason is that the average coccoliths length in the suspensions decreases slightly with settling time (see the Figure D1-c). But as proved in Appendix D, this variation won't draw significant influence on our velocity result. To be honest, we don't know the exact function neither know how to calculate it. In this study, we used a threshold of $R_{cal}$=15% to avoid variations in the average sinking velocity with coccolith size dynamics (this has been described in Appendix D of the former version). Only one data point of small *Ca. leptoporus* in our dataset was significant smaller than 15% (~5%). We think it is interesting to discuss the relationship between average sinking velocity and time, but this topic is beyond the scope of this study and perhaps also beyond our experiment conditions.

Your guess about a certain size of coccoliths vanishing from the upper column is correct and that's the principle of coccolith separation by settling method. We did not descript the protocol details because we do not present a fundamentally new protocol for separation in this study. If we know two coccoliths' sinking velocities and their difference is large enough, we can chose the settling duration easily by T=D/v, where v is the larger sinking velocity between the two kind of coccoliths. But as all reviewers' suggest, we have added this brief description in the last part of the main text.

For the Monte Carlo method here, we resampled the coccolith length from the assumed length distribution but this process is a little difference from typical Monte Carlo simulation. Because we only used the resampling dataset for a one-time simulation and did not repeat the simulation many times (we can do repeat simulations but the result can hardly fully plotted on this figure because of huge data amount). So, we have removed the term 'Monte Carlo' to avoid misleading readers. Moreover, we have added more descriptions for this simulation in Lines 537-540.

We have redrawn Figure D1 adding the fitting results in D1-b following your suggestion. We think the new figure can illustrate the statement 'we can assume the average sinking velocity as the sinking velocity of the the coccoliths with the average length' better. Thank you for this suggestion.

The points in Figure D1-b are what we measured in experiments and those in Figure D1-d are from simulations. We have explained this in Lines 561-563.

**Comments 14:** The ratio given in line 458 is not the number of coccoliths in a thickness dD as stated - as the authors have defined here, it is the number of coccoliths per unit unit thickness.

**Reply:** We have added a statement "dD is unit thickness".

**Comments 15:**

    a)  482 equation 2-6 doesn't exist. Should this be D-6?
    b)  eq. D-6 This is difficult to follow. Keep equation in symbol format before introducing numbers
    c)   eq. D-7 What is -10, and what is k?

**Reply:** Yes. Equation 2-6 should be D-6 and we have rewritten equation D-6 following your suggestion.

In equation D-7, 'k' is the slope of $R_{cal}$ against T. We defined it just above this equation in Line 547 (Line 484 in former version). If we use $V_1$=15 ml, $V_2$=10 ml and D=6 cm, the equation D-5 will be:

$$R = \frac{3}{5} - \frac{v}{10} \times t \qquad\qquad (eq. 1)$$

Here R is equal with $R_{cal}$, v is sink velocity and t is time. The slope of R-t, marked as k, is '-v/10'. This process was done just for a simplification of calculation and making our raw data more comparable and clearer as described in Line 539-544.

**Comments 16:**

Firstly, all coccoliths belong to a particular species are assumed to sink at exactly the same rate. Secondly, they are assumed to sink at a constant velocity from the instant that the suspension is left. I would like to see a calculation in the appendix estimating the time and distance that a particle falls before it reaches terminal sinking velocity, to show whether or not it is justifiable to ignore the accelerating phase for all of the particle sizes considered here. Intuitively I imagine this is a fair assumption, but it would be nice to see in numbers.

**Reply:**  We did not assume all coccoliths to sink at same rate. Our assumptions are two parts: (1) the sinking velocity we measured is the average sinking velocity of all coccoliths of a certain species; (2) the average sinking velocity can represent the sinking velocity of coccolith with a mean length for that species. This assumption has been stated in Lines 135-140. However, we failed to explain the proof clearly in Appendix D, so we have illustrated this in the **reply** to **Comments13** and improved it.

For your second questio, let us do some simple calculations to prove it. Because coccolith hydrodynamics is too complex to be calculated accurately, we take a calcite sphere as an example to show how fast can it reach terminal speed. Here we use the term 'terminal speed' to describe the speed when coccoliths sink in force balance.

If we chose downward force or speed as positive, the movement of a calcite sphere can be described by Newton's second law as following equation:

$$F = \frac{4}{3}\pi r^3 \rho_{cal}g \ - \ \frac{4}{3}\pi r^3 \rho_{water}g - 6\pi\eta rv = \frac{4}{3}\pi r^3 \rho_{cal}\frac{dv}{dt} \qquad \text{(eq. 2)}$$

Where F is the resultant of force, $r$ is sphere radium, $\rho_{cal}$ is the density of calcite (2.7 g cm$^{-3}$), $\rho_{water}$ is the density of water (~1.0 g cm$^{-3}$), $\eta$ is the viscosity of water, $v$ is sinking velocity of sphere. The second term of eq.2 is gravity, the third one is buoyancy, the next one is drag force and the term in the left of second equal sign is the sphere mass multiplied by accelerated speed. The eq. 1 can be modified to the following form:

$$\frac{dv}{dt} = \ -\frac{9\eta}{2r^2}v + \frac{g}{\rho_{cal}}(\rho_{cal} - \rho_{water}) \qquad \text{(eq. 3)}$$

We can simply the equation as following:

$$\frac{dv}{dt} = \ av + b \qquad \text{(eq. 4)}$$

where a, b and c are as following

$$a = -\frac{9\eta}{2r^2} \qquad \text{(eq. 5)}$$

$$b = \frac{g}{\rho_{cal}}(\rho_{cal} - \rho_{water}) \qquad \text{(eq. 6)}$$

$$c = \ln b \qquad \text{(eq. 7)}$$

Solve the differential equations with an initial value $v_{t=0}=0$, we can get:

$$v = \frac{e^{(c+at)} - b}{a} \qquad \text{(eq. 8)}$$

So the sinking velocity, v, as a function of sinking time, t, can be written as following equation:

$$v = \frac{-e^{[-\frac{9\eta}{2r^2}t+\ln(-\frac{g}{\rho_{cal}}(\rho_{cal}-\rho_{water}))]}+\frac{g}{\rho_{cal}}(\rho_{cal}-\rho_{water})}{\frac{9\eta}{2r^2}} \qquad \text{(eq. 9)}$$

Ignoring other parameters, if we set the time, 't', to large enough (or we can say infinite mathematically), we can get the terminal speed (marked as $v_t$), which is exactly same as the Stocks' law:

$$\lim_{t\to\infty} v = \frac{2(\rho_{cal}-\rho_{water})gr^2}{9\eta} \qquad \text{(eq. 10)}$$

But actually, $v$ can equal to $v_t$ even when t is a quite small number. We can see the term, $e^{(c+at)}$, in eq. 7 will be close to zero when a*t is negative shifting. If we set $r$ varies between 1*10$^{-6}$m to 1*10$^{-5}$m (typical coccolith size), $a$ will be ~-10$^9$, while c is only about 1.8. As long as, $t$ is close to 10$^{-7}$ s, the exponent term will be almost close to zero (e.g. exp(-10$^2$)=3.7*10$^{-44}$) making the sinking velocity equals to balance velocity. This value (t=10$^{-7}$ s) is about 11-12 order of magnitude smaller than the time we discuss in our paper. So the assumption that coccolith can reach the terminal speed fast is reasonable. We believe that it is not essential to include the above derivation in the manuscript, following the articles about particles settling cited in our manuscript.

**Comments 17:**

I would like a more in depth discussion of these features and other factors affecting sinking velocities in the lab - for example - temperature gradients leading to convection, entrainment of small particles by larger ones (i.e. do smaller coccoliths sink faster when there are large coccoliths present?).

**Reply:** Good suggestions. We never considered the convection caused by temperature gradients. Because one of the foundations of this experiment is all coccoliths sinking velocities are in still solutions. In settling, there is no temperature gradient and no evidence for convection. Because the solution temperature is homogeneous and constant during the experiment.

There has been a lot of papers discussing a multi-species particles in hindering settling. In Masliyah's calculation (1979), the velocities of smaller particles only decrease significant when the volume of particles excess 10%. In our experiments, the volume of sediments are controlled below 5%. And there is another study calculating the different size particles with same density in a hindering settling process (Greenspan and Ungarish, 1982). However, we think such a discussion is beyond our study's scope.

[revised manuscript text omitted]

R and (c) and (d) represent the absolute error of R.

---

## Author Comment (AC3) · 21 Apr 2018

We thank the reviewer for these constructive suggestions.

**Comments 1:**

The authors took most of the paragraphs to describe and calculate the sinking velocities of different coccolith species. However, what is the application of this parameter in future research? This is not very clear to me. I think the purpose of this paper is to give the audience "a refinement of coccolith separation method". So I suggest adding some paragraph to introduce how to use your SV data in routine work or to give the audience some suggestions how to improve the efficiency or precision of the separation method after your work.

**Reply:** In this study, we indeed focused on the measurement of coccolith sinking speeds. We do not try to propose a new protocol for coccolith separation, instead our empirical data can be used to refine settling time choices using existing protocols. Once sinking velocities are estimated for coccoliths in a particular sample set, coccoliths can be separated by the protocols described in Bolton et al. (2012) or Stoll and Ziveri (2002), using optimal settling times, vessels, and concentrations from this study. We have added a brief descriptions of separation protocol (Lines 247-272).

**Comments 2:**

In this manuscript, the authors used several technique and methods in the experiments, such as "sinking method or filtering method" in L83 or "drop technique" in L99. This would be difficult to follow for the audience who are not very familiar with coccolith separation. I suggest adding some brief explanations of these techniques.

**Reply:** Thank you for this suggestion. We have added a brief description of the micro-filtering method in Lines 41-43. The description of sinking/decanting method can be found in the Line 49-53. For the drop technique, we have rewritten chapter 2.2.2 and added more details about this method (Lines 101-119).

**Comments 3:**

The authors selected eight raw sediment samples from different cores in global oceans. As I know, these cores have different geographic settings like different water depths, mineral composition and nannofossil preservation. Do these factors influence the separation process or the sinking velocity?

**Reply:** Yes, these factors may influence the separation process. We have inclued a short discussion of the potential influence of dissolution on sinking velocity in Lines 233-235 (Lines 206-207 in former version). But we do not discuss the influence of thickness on coccolith sinking velocity, which will be an interesting point for future study. As for the mineral composition, we suggeste that if the content of suspension is below a certain level, the clay or quartz or any other mineral particles can be ignored in hindering settling (similar question was also answered in **Reply** to Review 1's **Comments 1**).

**Comments 4:**

The section of "Conclusions", this part is more or less like a part of discussion and not so constructive to me. I suggest improving this part.

**Reply:** We have rewritten this part as 'Conclusion and suggestion for separation'. See the new version.

**Comments 5:**

In L86-87, "except the Pseudoemiliania lacunosa and Umbilicosphaera sibogae, which cannot be separated from each other". Why? Should give some explanations.

**Reply:** The only reason is they have similar sinking velocities. We have explained this in our new 'Conclusion and suggestion for separation' section.

**Comments 6:**

In L188-189, "If we use data for all species except Helicosphaera carteri. . ." why don't include H. carteri in the calibration?

**Reply:** Thanks for pointing out that this needs clarification. We didn't use *H. carteri* in the regression because of its specific shape, which is quite different to the other species studied. This was explained it in Lines 224-233 (Lines 203-205 in the former version): the ellipticity of *H. carteri* (~0.6) is significant lower than other coccolith (among 0.8-0.9), therefore its settling behavior differs from other species. This is also illustrated in Figure 6d and Figure C3. We have reorganized this part to make it clearer.

**Comments 7:**

L66, change "two Neogene samples" to "two Neogene/Paleogene samples"

**Reply:** Done.

**References**

[revised manuscript text omitted]

R and (c) and (d) represent the absolute error of R.

---

## Author Response (AR3)

Dear the Editor,

Thank you for your and reviewer's suggestion. The details of revision are listed below.

For example, line 8 and 39: there is an extra space here. Please remove and check the rest of the text.

**Reply:** Done.

line 40: 'mirco' should be 'micro'

**Reply:** Done.

line 137: 'In about 500 ml diluted ammonia.'

**Reply:** Done.

line 217: remove one of the dots.

**Reply:** Done.

line 243 and elsewhere: should be 'C. leptoporus' rather than 'Ca. leptoporus'.

**Reply:** Done. We also change the 'Cy. floridanus' to 'C. floridanus' and 'Co. pelagicus' to 'C. pelagicus'

line 234: perhaps add Tremblin et al. 2016 - PNAS in the papers using the micro filtering technique.

Answer: The method of Tremblin 's research was same as Minoletti et al., which has been cited in the Introduction part.

Regarding equation 2-2: could you take a step towards isolating the effects of particle size, particle shape, particle material density, vessel shape, suspension density... etc? I don't see this as essential to the present manuscript, if the intention is purely to improve the protocol.

**Reply:** We think the equation 2-2 is for describing the method, so perhaps it's better to keep it simple. We had shown the equation special for the shape, particle density and suspension in other equations in the 'Discussions' part and made a summary in the 'Suggestions' part.

Line 150: This currently reads as though the vessel is smaller than the particle, and thus doesn't make sense.

"A significant wall effect will be detected when a particle is settling in a vessel which diameter is smaller than the particle size by two orders of magnitude (Barnea and Mizarchi, 1973)"

**Reply:** Yes, we realized this is an ambiguous sentence. We changed it as "...when a particle is settling in a vessel with a diameter that is smaller than **100 times of the particle size**". Thank you for pointing it out.

Regarding comment 9 by reviewer #2: although a Monte Carlo approach is indeed common, it is a rather opaque method of estimating uncertainty when the exact inputs are not specified. The uncertainty in a value is quite as important as it's expected value so it is therefore important to be clear about what goes into this analysis. The author's explanation here may be acceptable, but this does not mean that details of their approach should be omitted. A description of the above should at least go into the appendix for anyone who wants to reproduce their results.

**Reply:** We think we have offer enough information for reproducing the error estimation. Because the Poisson distribution is different from the normal distribution: the exception is equal to the variance. That means we don't have to describe the specified inputs such as the variance, which is a quite important parameter for other methods. We listed the name of matlab functions in the new version in the line 573-574 to make it easier to reproduce the results.

Regarding comment 14 by reviewer #2: the ratio: "Nu(t=0) / D" is the number of coccoliths per unit thickness. dD doesn't come into it as you're not taking a derivative with respect to D. Please adjust.

**Reply:** We had replaced the 'dD' by unit thickness, so there is no 'dD' in the current version.

The description in appendix D is much improved. The assumptions have been stated more clearly than in the first version. One point however is that when equation D-1 is presented, it should be made clear that either A) as written this is for a coccolith of a single size, with all coccoliths sinking at identical rates. Or B) that the variables N_u and v are actually distributions. It is not obvious a priori how these distributions will change throughout the course of settling and thus whether it's valid to use the mean of the distribution for all calculations.

**Reply:** We added the assumption: the velocity is the average sinking velocity. This assumption will be proved in the following (line 504-506).

It is appreciated that the authors due diligence showing that constant velocity throughout settling is reasonable. This is useful and should at least be stated explicitly as a reasonable assumption.

**Reply:** We added this reason of this assumption in the line 506-508 without any detail calculations.

[revised manuscript text omitted]